# Correlation between Cross-Sectional Anatomy and Computed Tomography of the Normal Six-Banded Armadillo (*Euphractus sexcintus*) Nasal Cavity and Paranasal Sinuses

**DOI:** 10.3390/ani14071135

**Published:** 2024-04-08

**Authors:** José Raduan Jaber, Daniel Morales Bordon, Alberto Arencibia, Juan Alberto Corbera, Magnolia Conde-Felipe, Maria Dolores Ayala, Mario Encinoso

**Affiliations:** 1Departamento de Morfología, Facultad de Veterinaria, Universidad de Las Palmas de Gran Canaria, Trasmontaña, Arucas, 35413 Las Palmas, Spain; alberto.arencibia@ulpgc.es; 2Departamento de Patología Animal, Producción Animal, Bromatología y Tecnología de Los Alimentos, Facultad de Veterinaria, Universidad de Las Palmas de Gran Canaria, Trasmontaña, Arucas, 35413 Las Palmas, Spainmagnolia.conde@ulpgc.es (M.C.-F.); 3Hospital Veterinario, Facultad de Veterinaria, Universidad de Las Palmas de Gran Canaria, Trasmontaña, Arucas, 35413 Las Palmas, Spain; mencinoso@gmail.com; 4Departamento de Anatomía y Anatomía Patológica Comparadas, Facultad de Veterinaria, Universidad de Murcia, 30100 Murcia, Spain

**Keywords:** cross sections, computed tomography, anatomy, nasal cavity, paranasal sinuses, head, six-banded armadillo

## Abstract

**Simple Summary:**

The six-banded armadillo (*Euphractus sexcintus*) is included in the International Union for Conservation of Nature (IUCN) as being of the least concern, mainly due to its wide distribution and large population. Nonetheless, in recent years, its population is deploying in specific areas as a consequence of human activities. The scarce literature regarding the anatomy of this species leads us to investigate the normal six-banded armadillo sinuses and nasal passages using modern imaging techniques such as computed tomography (CT) combined with anatomical cross-sectioning to acquire helpful information on the structures that comprise this region.

**Abstract:**

This research aimed to study the rostral part of the head of the six-banded armadillo, applying advanced imaging techniques such as CT. Furthermore, by combining the images obtained through this technique with anatomical cross-sections, an adequate description of the structures that constitute the rostral part of the head of this species is presented. This anatomical information could provide a valuable diagnostic tool for the clinical evaluation of different disorders in the six-banded armadillo’s nasal cavity and paranasal sinuses.

## 1. Introduction

The six-banded armadillo (*Euphractus sexcintus*) is a unique member of the genus *Euphractus*, belonging to the Chlamyphoridae family [1]. This animal shows a wide geographical distribution in South America east of the Andes, from Brazil and southern Suriname in the northeast through Bolivia, Paraguay and Uruguay into northern Argentina in the southeast [2,3]. They inhabit dry savannahs or parts of wet savannahs, where armadillos build burrows with a single inverted U-shaped entrance [2]. The generic name of the species comes from their ossified dermal plates of equal lengths being arranged in a specific pattern. These plates protect the top of the head, flanks, back, and tail, and are sparsely covered by thin grey-brown hair [2]. On the back, they form a carapace that is a pale yellow, pale tan, or reddish tan [2]. This armadillo has a gestational period of approximately two months, giving birth to a litter that ranges from one to three baby six-banded armadillos of both sexes [1,2,3]. At this time, if the females are disturbed, they try to withhold themselves and their young from sight or move the young and behave vigorously against the disruption. These babies reach maturity around nine months of age [2]. Nonetheless, this species is extremely timid and prefers to escape predators rather than fight [3].

Like all *Euphractus*, this species is diurnal, although some researchers have recorded important activity at night [4]. Their diet is omnivorous, consuming a wide variety of animals and plants, including carrion, small vertebrates, insects such as ants and termites, and plant material [2,3,4]. This material (bromeliad fruit, tubers, palm nuts) represents a relevant proportion of their diet, as pointed out by Schaller [4]. Moreover, the six-banded armadillo can gather a considerable amount of subcutaneous fat as a consequence of a seasonal insufficiency in food [5]. Despite the fact that the IUCN does not give this animal any special status as a threatened species, it is pivotal to highlight that their population is declining due to there being few patches of savannas, and human interactions [3]. 

Concerning their main anatomical remarks, they have five complete toes that end in unmodified claws that are notably developed for digging and the construction of burrows [2,3]. The head is broad and flattened, with the facial region showing a triangular-like shape. The frontal bone is well developed, and the orbital fossa presents a lateral disposition. Other additional features include a slender and elongated mandible with large and strong teeth and powerful muscles for mastication [2,6].

The great anatomical complexity between mammalian species and the growing interest in wildlife and exotic animals as companion animals have represented a challenge to veterinary clinicians in the diagnostic imaging studies interpretation [7]. Imaging diagnostic techniques have brought a notable advance in their clinical practice due to the facility to get essential information on the organization that constitutes the animal body [8]. Due to technological advances in this field, anatomical and diagnostic information can be obtained in an easy and considerably faster way, becoming a remarkable tool for this practice, using traditional procedures, such as radiology and ultrasounds [7], and modern imaging diagnostic techniques such as CT or magnetic resonance imaging (MRI) [8,9,10,11,12,13,14], which display better imaging resolutions, fast image acquisition, and overlapping structures’ avoidance, among others. Therefore, these techniques provide anatomical and functional information with a greater ability to distinguish between bony and soft tissue structures.

Some remarkable investigations on the anatomy, physiology, and pathology of these species are already available, mainly focused on the nine-banded armadillo [15,16,17,18,19]. Therefore, authors have reported the anatomy and functional morphology of xenarthrous vertebrae [15], the development of the spine [16], the study of the skeleton and the skull [6,17,18,19], the description of the nasal cavity and paranasal sinuses [20,21], the morphology of the laryngeal cartilages [22], the functional assessment of masticatory adaptations [23], as well as pathological processes, including fractures [24], and osteoderm lesions [25]. However, to the best of the authors’ knowledge, no studies including the nasal cavity and paranasal sinuses of the six-banded armadillo have been conducted [6]. Hence, this investigation aimed to describe the normal anatomy of the sinuses and nasal passages of the six-banded armadillo using CT, and those anatomic sections that can contribute to understanding all the structures that comprise the rostral part of the head of this species. 

## 2. Materials and Methods

### 2.1. Animals

Three carcasses (two males and one female) of adult six-banded armadillos (*Euphractus sexcintus*) from the zoological park called “Rancho Texas Lanzarote Park” (Lanzarote, Canary Islands, Spain) were collected. The animals died due to natural causes. 

### 2.2. CT Technique

To obtain CT images, a 16-slice helical CT scanner (Toshiba Astelion, Canon Medical System, Tokyo, Japan) was used at the Veterinary Hospital of Las Palmas de Gran Canaria University. The armadillos were placed in ventral recumbency in a symmetrical position on the CT scan table (Figure 1). Sequential CT images with a thickness of 1 mm were obtained using a standard clinical protocol (100 kVp, 80 mA, 512 × 512 acquisition matrix, 1809 × 858 field of view, a spiral pitch factor of 0.94, and a gantry rotation of 1.5 s). To optimize the CT appearance of the nasal cavity and paranasal sinuses, three CT windows with different widths and levels were used: a bone window setting (WW = 1500; WL = 300), a lung window setting (WW = 1400; WL = −500), and a soft tissue window setting (WW = 350; WL = 40). Moreover, dorsal and sagittal multiplanar reconstructed (MPR) images were also obtained to improve the visualization of normal armadillo anatomical structures. All the CT images were uploaded to an image viewer (OsiriX MD, Apple, Cupertino, CA, USA) for data manipulation and analysis.

### 2.3. Anatomic Evaluation

Anatomical cross-sections were obtained to facilitate the identification of the structures identified in the CT images. After the scanning procedures, these specimens were placed in a plastic isolation holder in ventral recumbency and successively stored in a freezer (−80 °C) until completely frozen. Subsequently, the three frozen carcasses were sectioned using an electric band saw to obtain sequential cross-sections. Contiguous 1 cm transverse slices were obtained starting at the nose and extending to the ethmoidal bone. These slices were thicker than those of the CT to preserve the integrity and position of the anatomic structures in the sections. These sections were cleaned with water, numbered and photographed on the cranial and caudal surfaces. Afterwards, the anatomic sections that better matched the CT images to identify the significant structures of the six-banded armadillo’s sinuses and nasal passages were selected. Additionally, anatomical texts and notable references describing these species were also used [4,19,20,21,22,23,24,25,26,27,28,29,30,31,32,33,34]. 

## 3. Results

No anatomic variations were detected in the head of the six-banded armadillos used in the investigation. Here, different sections revealing the main anatomical formations of the six-banded armadillo nasal cavity and paranasal passages are displayed (Figure 2, Figure 3, Figure 4, Figure 5, Figure 6, Figure 7, Figure 8, Figure 9 and Figure 10). Figure 2 corresponds with a sagittal CT image, in which each line and number (I–VII) represent approximately the level of the following anatomical and transverse CT images. Figure 2, Figure 3, Figure 4, Figure 5, Figure 6, Figure 7, Figure 8 and Figure 9 are composed of four images: (A) an anatomical cross-section, (B) a bone CT window image, (C) a soft tissue CT window image, and (D) a lung CT window image. The images are presented in a rostrocaudal progression from the nose (Figure 3) to the ethmoidal labyrinth (Figure 9). Finally, Figure 10 is composed of three CT images: (A) a parasagittal MPR CT image and (B,C) dorsal MPR CT images at the level of the ventral nasal concha and dorsal nasal concha, respectively.

### 3.1. Anatomical Sections

Clinically relevant structures of this region were identified via anatomical cross-sections. Thus, the selected images showed most of the structures of the nasal cavity and paranasal sinuses. 

Thus, the rostral opening of the nasal cavity represented by two symmetrical apertures called nostrils, which had circular shapes, was depicted. The outer more-or-less flaring wall of each nostril is the Radix nasi which extends rostroventrally as the Dorsum nasi till finished in the Apex nasi (Figure 3A). More caudally, the median nasal septum is distinguished, covered by a mucous membrane, and attached dorsally to the nasal bone (Figure 4A). This structure divides the nasal cavity into two symmetrical halves. This nasal septum extended dorso and ventrolaterally, forming the dorso and ventrolateral nasal cartilages on both sides, respectively (Figure 4A). The transverse images were crucial for visualizing the alar, straight, and basal folds (Figure 4A and Figure 5A). This opening widened caudally to be part of the entrance of the nasal cavity. Into this cavity, we observed the nasal conchae projected into each half of the nasal cavity. We could distinguish the dorsal and ventral nasal conchae (Figure 6A, Figure 7A and Figure 8A). In each nasal passage, the shelf-like dorsal nasal concha and the scrolls of the ventral nasal concha divide the nasal cavity distinguishing four different passages or meatuses. Therefore, we identified the dorsal nasal meatus that lays between the nasal bone and the dorsal nasal concha (Figure 5A, Figure 6A, Figure 7A and Figure 8A), the middle nasal meatus lying between the dorsal and ventral nasal conchae (Figure 7A and Figure 8A), and the ventral nasal meatus that was depicted dorsally to the hard palate (Figure 5A, Figure 6A, Figure 7A and Figure 8A). Moreover, we could observe the common nasal meatus or the space between the conchae and both sides of the nasal septum (Figure 7A and Figure 8A and Figure 9A). The fundus of the nasal cavity is occupied by the ethmoidal labyrinth (Figure 9A) and is composed of several delicate scrolls attached to the cribiform or horizontal plate caudally (Figure 10A). Moreover, these images distinguished the opening of the maxillary sinus into the nasopharynx (Figure 9A). 

Additionally, these sections allowed the identification of different structures of the oral cavity such as the tongue and the oral vestibule that comprises the cavity lying outside the teeth and gums (Figure 6A, Figure 7A, Figure 8A and Figure 9A). Adjacent structures such as the hard palate, the palatine plexus, and the nasopharynx were also well identified (Figure 7A and Figure 8A and Figure 9A). Furthermore, pivotal bones related to the nasal cavity and paranasal sinuses, including the incisive, nasal, palatine, and maxilla were also identified (Figure 3A, Figure 4A, Figure 5A, Figure 6A, Figure 7A, Figure 8A and Figure 9A). Ventrally, the mandible bearing the lower teeth (Figure 4A, Figure 5A, Figure 6A, Figure 7A, Figure 8A and Figure 9A), as well as other muscles, including the *buccinator*, *geniohyoideus*, and *hyoglossus* muscles, were distinguished (Figure 6A, Figure 7A, Figure 8A and Figure 9A). 

### 3.2. Computed Tomography (CT)

Regarding the CT images, they were quite helpful in distinguishing the relevant structures that comprise the nasal cavity and paranasal sinuses. Thus, all the CT windows provided adequate details of the nostrils showing a round shape, mainly in the soft tissue CT window (Figure 3B–D). Moreover, the lung and bone CT windows displayed the philtrum with notable detail. Therefore, it was distinguished as a Y-shape cleft in the lower third of the face (Figure 3C,D). Dorsally to the nostrils, we could visualize a hyperattenuated structure that corresponded with the rostral part of the nasal bone (Figure 3B–D). Other remarkable structures such as the nasal septum were depicted with all the CT windows used, displaying heterogeneous attenuation (Figure 4B–D, Figure 5B–D, Figure 6B–D, Figure 7B–D and Figure 8B–D). This was composed of cartilaginous and bony parts. This osseous part is more caudal and formed, among others, by the perpendicular and horizontal lamina of the ethmoidal bone and the vomer (Figure 9B–D and Figure 10B,C). The lung and bone tissue windows were essential in distinguishing the different folds, as well as the dorsal, middle, ventral, and common nasal meatuses (Figure 4B–D, Figure 5B–D, Figure 6B–D, Figure 7B–D, Figure 8B–D and Figure 9B–D and Figure 10A–C). More caudally, it was possible to identify the ethmoidal bone complex that is located between the braincase and the facial part of the skull. This complex was well visualized with the different windows employed (Figure 9B–D and Figure 10A–C), which consisted of the ethmoidal labyrinth, the cribiform or horizontal lamina, and the median bony perpendicular lamina of the nasal septum. Additionally, the lung and bone CT tissue windows were adequate in revealing the organization of this labyrinth. Therefore, the ectoturbinates were distinguished to be extending dorsally (illustrated in Figure 9C,D), and close to the frontal sinus (Figure 8B,D and Figure 10A), whereas the ectoturbinates could be identified as being attached to the vomer, which could serve as a reference to separate the ethmoidal labyrinth from the nasopharynx (illustrated in Figure 9C,D). In addition, the transverse lung and bone CT images depicted the communication between the maxillary sinus and the nasopharynx (Figure 9C,D). 

The transversal, sagittal, and dorsal CT images allowed for the identification of bone structures related to the nasal cavity. Subsequently, the nasal, incisive, maxilla, frontal, pterygoid, and palatine bones were distinguished (Figure 4B–D, Figure 9B–D and Figure 10A–C). Moreover, other bony structures such as the mandible and the hyoid apparatus were also identified (Figure 4B–D, Figure 9B–D and Figure 10A). Concerning those structures with an intraluminal gas content, such as the nasal conchal recess, oral cavity, nasopharyngeal duct, as well as the maxillary (divided into rostral and caudal parts), and the sphenoidal and frontal sinuses (illustrated in Figure 8B,D, Figure 9C,D, and Figure 10A–C), could be identified with this technique appearing with a vacuum effect. In addition, there were areas of soft tissue attenuation lateral to the mandible and bilateral to the hyoid apparatus that were compatible with different muscles such as the *genioglossus*, *geniohyoideus*, and *buccinator* muscles (Figure 6B–D, Figure 7B–D, Figure 8B–D and Figure 9B–D).

**Figure 3 animals-14-01135-f003:**
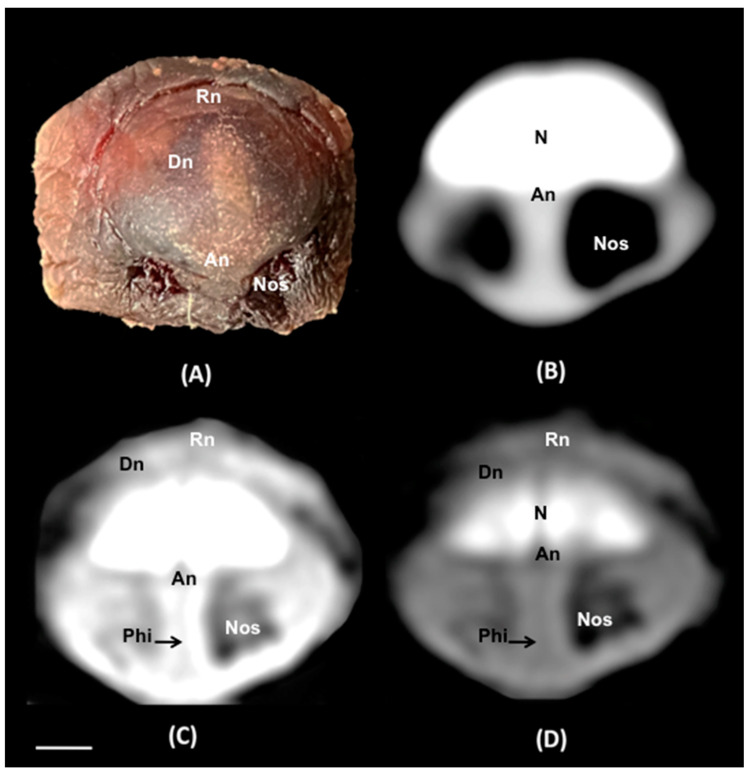
Transverse cross-section (**A**), soft tissue (**B**), lung (**C**), and bone (**D**) CT images of the nasal cavity of a six-banded armadillo’s nasal cavity at the level of the nose corresponding to line I in Figure 2. N: nasal bone. Nos: nostrils. Dn: *dorsum nasi*. An: *apex nasi*. Rn: *radix nasi*. Phi: *Philtrum*. Scale bar: 1 cm.

**Figure 4 animals-14-01135-f004:**
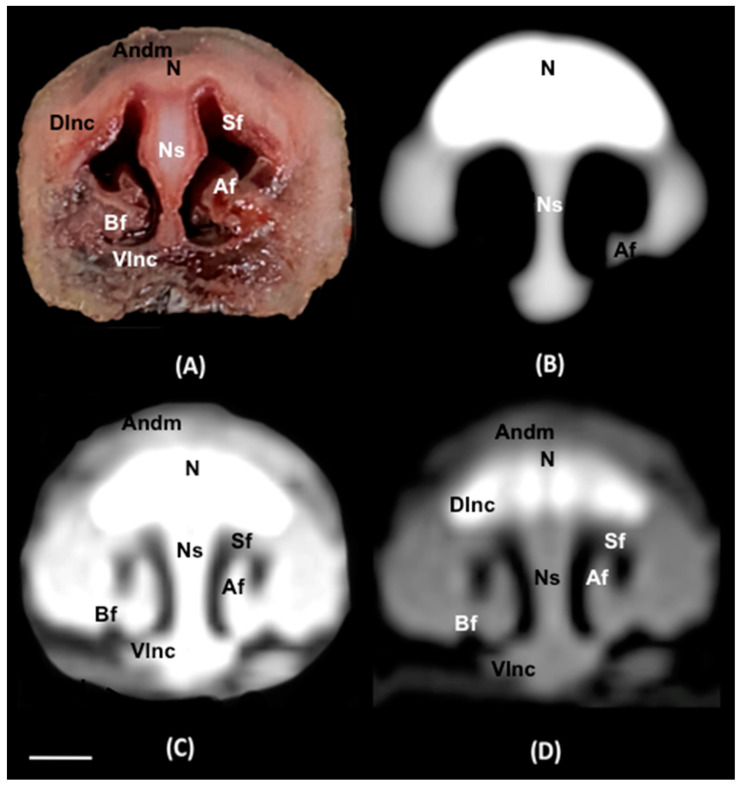
Transverse cross-section (**A**), soft tissue (**B**), lung (**C**), and bone (**D**) CT images of the nasal cavity of a six-banded armadillo’s nasal cavity at the level of the alar fold corresponding to line II in Figure 2. Andm: apical naris dilatator muscle. N: nasal bone. Ns: nasal septum (cartilage). Sf: straight fold. Af: Alar fold. Bf: basal fold. Dlnc: dorsolateral nasal cartilage. Vlnc: ventrolateral nasal cartilage. Scale bar: 1 cm.

**Figure 5 animals-14-01135-f005:**
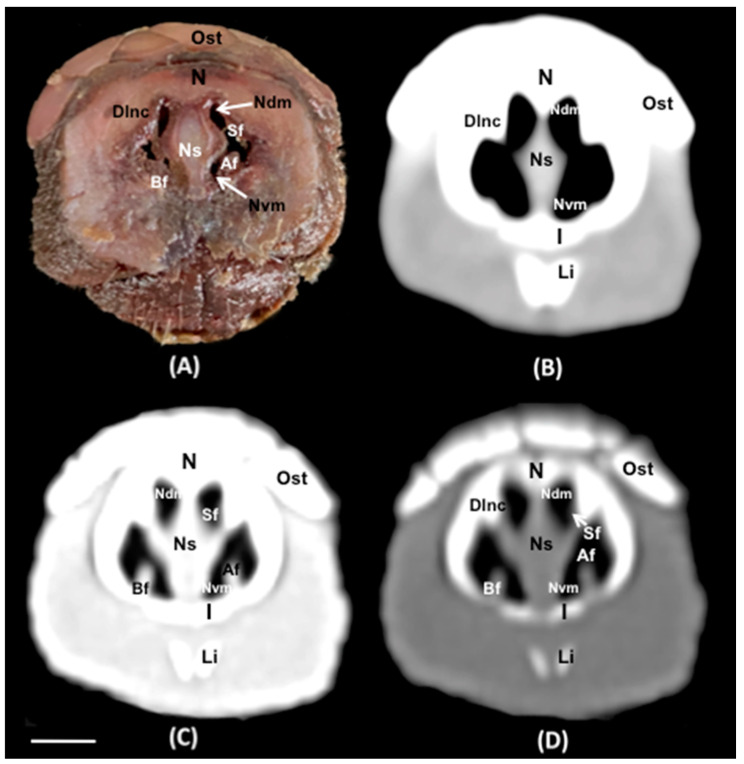
Transverse cross-section (**A**), soft tissue (**B**), lung (**C**), and bone (**D**) CT images of the nasal cavity of a six-banded armadillo’s nasal cavity at the level of the dorsolateral nasal cartilage corresponding to line III in Figure 2. Ost: osteoderm. N: nasal bone. Ns: nasal septum (cartilage). Sf: straight fold. Af: Alar fold. Bf: basal fold. Dlnc: dorsolateral nasal cartilage. Ndm: nasal dorsal meatus. Nvm: nasal ventral meatus. I: incisive bone. Li: lower incisor. Scale bar: 1 cm.

**Figure 6 animals-14-01135-f006:**
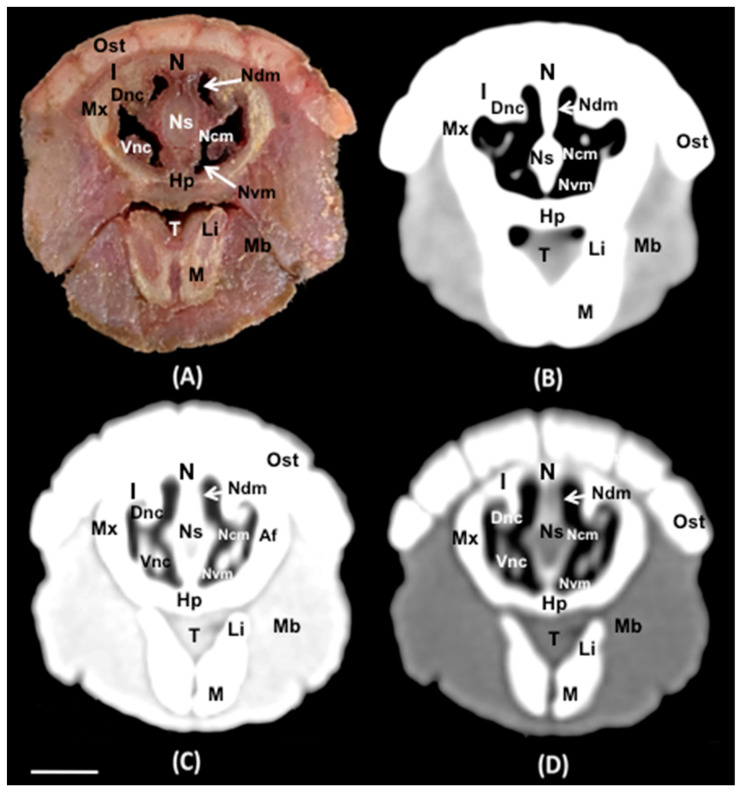
Transverse cross-section (**A**), soft tissue (**B**), lung (**C**), and bone (**D**) CT images of the nasal cavity of a six-banded armadillo’s nasal cavity at the level of the incisive bone corresponding to line III in Figure 2. Ost: osteoderm. N: nasal bone. I: incisive bone. Mx: Maxilla. Ns: nasal septum (cartilage). Dnc: dorsal nasal concha. Vnc: ventral nasal concha. Ndm: nasal dorsal meatus. Ncm: nasal common meatus. Nvm: nasal ventral meatus. Hp: hard palate T: tongue. Li: lower incisor. M: mandible. Mb: M. Buccinator (pars buccalis). Scale bar: 1 cm.

**Figure 7 animals-14-01135-f007:**
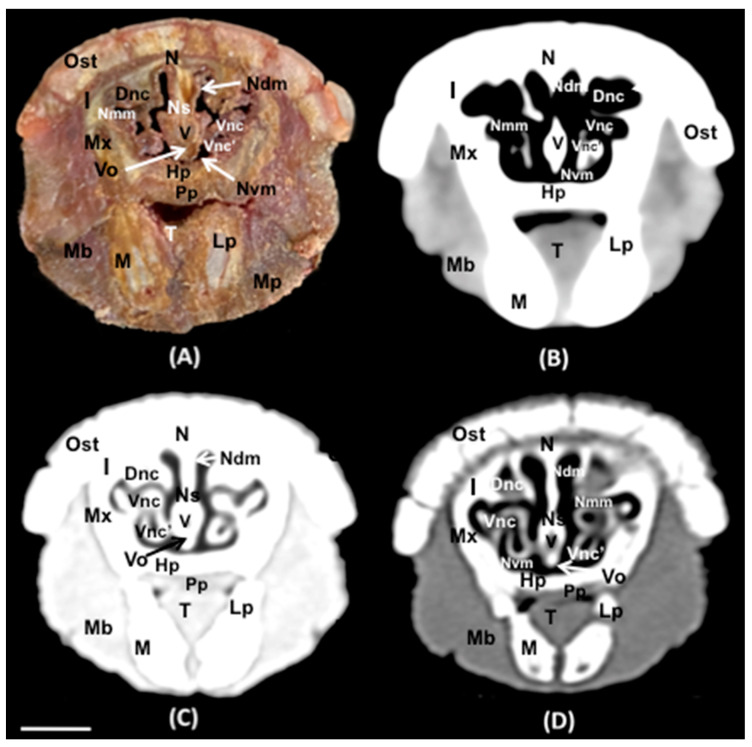
Transverse cross-section (**A**), soft tissue (**B**), lung (**C**), and bone (**D**) CT images of the nasal cavity of a six-banded armadillo’s nasal cavity at the level of the nasal conchae corresponding to line IV in Figure 2. Ost: osteoderm. N: nasal bone. I: incisive bone. Mx: Maxilla. Ns: nasal septum. Dnc: dorsal nasal concha. Vnc: ventral nasal concha (dorsal part). Vnc’: ventral nasal concha (ventral part). Ndm: nasal dorsal meatus. Nmm: nasal middle meatus. Nvm: nasal ventral meatus. V: vomer. Vo: vomeronasal organ. Hp: hard palate. Pp: Palatine plexus. T: tongue. Lp: lower premolar. M: mandible. Mb: M. Buccinator (pars buccalis). Scale bar: 1 cm.

**Figure 8 animals-14-01135-f008:**
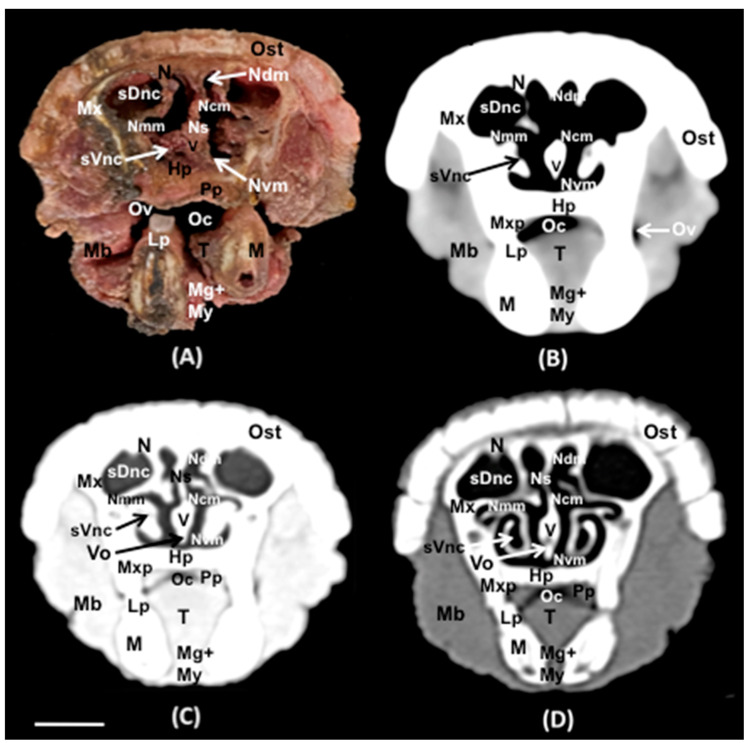
Transverse cross-section (**A**), soft tissue (**B**), lung (**C**), and bone (**D**) CT images of the nasal cavity of a six-banded armadillo’s nasal cavity at the level of the M. Geniohyoideus corresponding to line V in Figure 2. Ost: osteoderm. N: nasal bone (ethmoidal crest). Mx: Maxilla. Ns: nasal septum (cartilage). sDnc: sinus of dorsal nasal concha. sVnc: sinus of the ventral nasal concha. Ndm: nasal dorsal meatus. Nmm: nasal middle meatus. Nvm: nasal ventral meatus. Ncm: nasal common meatus. V: vomer. Vo: vomeronasal organ. Hp: hard palate. Pp: Palatine plexus. T: tongue. Oc: oral cavity. Ov: oral vestibule. Mxp: maxillar premolar. Lp: lower premolar. M: mandible. Mb: M. Buccinator (pars buccalis). Mg + My: M. Geniohyoideus + My. Mylohyoideus. Scale bar: 1 cm.

**Figure 9 animals-14-01135-f009:**
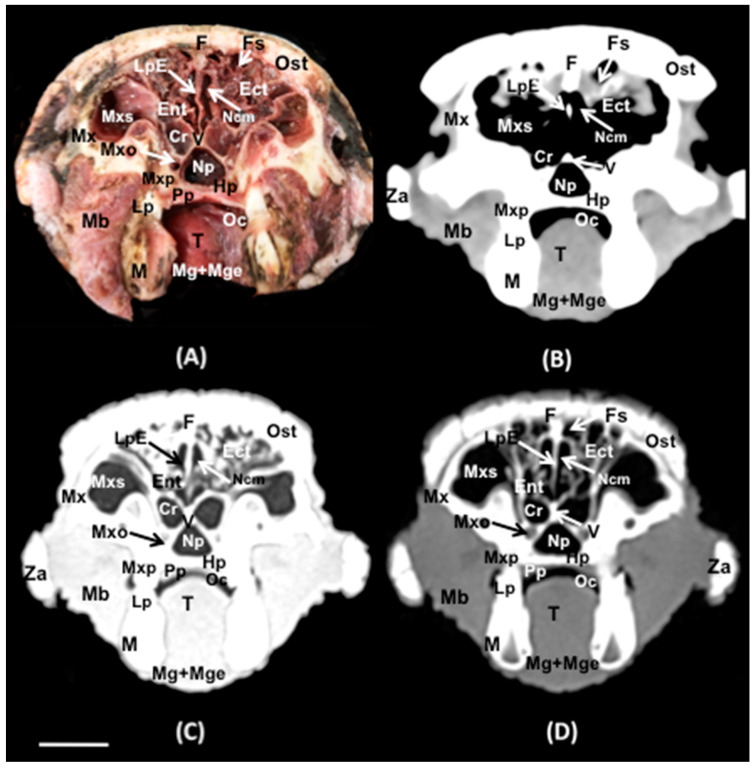
Transverse cross-section (**A**), soft tissue (**B**), lung (**C**), and bone (**D**) CT images of the nasal cavity of a six-banded armadillo’s nasal cavity at the level of the ethmoturbinates corresponding to line VI in Figure 2. Ost: osteoderm. F: frontal bone. Fs: frontal sinus. Mx: Maxilla. Ncm: nasal common meatus. Hp: hard palate. Pp: Palatine plexus T: tongue. Oc: oral cavity. LpE: Perpendicular lamina of the ethmoid bone. Ect: ectoturbinates. Ent: endoturbinates. Mxs: maxillary sinus. Mxo: maxillary opening. Cr: caudal recess of the nasal cavity (sphenoid sinus). V: vomer. Np: nasopharynx. Mxp: maxillar premolar. Lp: lower premolar. M: mandible. Za: zygomatic arch. Mb: M. Buccinator (pars buccalis). Mg + Mge: M. Geniohyoideus + M. Genioglossus. Scale bar: 1 cm.

**Figure 10 animals-14-01135-f010:**
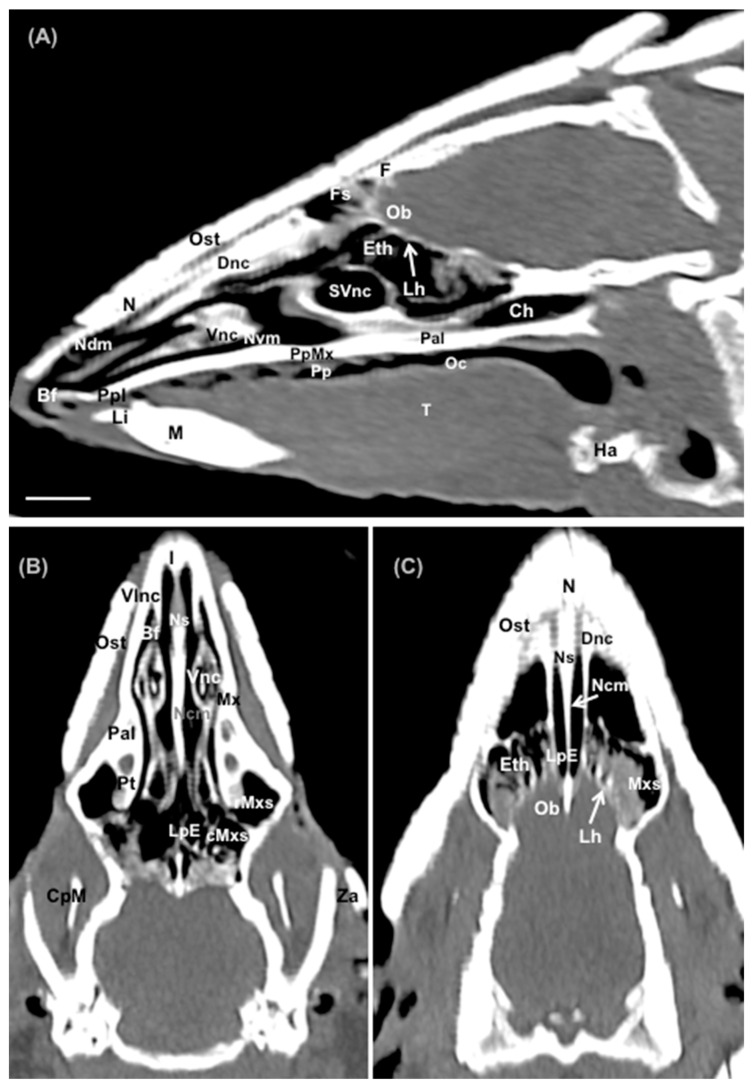
Parasagittal (**A**) and dorsal (**B**,**C**) CT images in soft tissue window of the nasal cavity of a six-banded armadillo’s head. Ost: osteoderm. I: incisive bone. N: nasal bone. F: frontal bone. Fs: frontal sinus. Ob: olfactory bulb. Mx: Maxilla. Dnc: dorsal nasal concha. Vnc: ventral nasal concha. sVnc: sinus of the ventral nasal concha. Ndm: nasal dorsal meatus. Nvm: nasal ventral meatus. Ncm: nasal common meatus. Vlnc: ventrolateral nasal cartilage. Bf: basal fold. PpI: palatine process of the incisive bone. PpMx: palatine process of the maxilla. Pp: palatine plexus. Pal: palatine bone. Pt: pterygoid bone. T: tongue. Oc: oral cavity. LpE: perpendicular lamina of the ethmoid bone. Lh: horizontal lamina of the ethmoid bone. Eth: ethmoturbinates. Ch: choana. Mxs: maxillary sinus. cMxs: caudal maxillary sinus. rMxs: rostral maxillary sinus. Li: lower incisor. M: mandible. CpM: coronoid process of the mandible. Za: zygomatic arch. Ha: hyoid apparatus.

## 4. Discussion

Modern imaging diagnostic techniques have improved anatomical knowledge and the recognition of different pathologies in veterinary medicine [6,7,8,9,10,11,12,13,14]. Compared to traditional procedures, such as radiology and ultrasounds, advanced imaging techniques provide excellent resolutions of anatomical formations, better definitions of the extent and character of lesions, fast imaging acquisition and the absence of superimposition, which have meant a transformation in research, veterinary practices, and teaching purposes [6,7,8,9,10,11,12,13,14,17]. However, the use of these techniques in exotic animal medicine is currently limited because of their expenses, availability, and the logistic problems of acquiring images of some of these species. Although descriptions of the CT and MRI of the nasal cavities and frontal sinuses have been done in exotic and traditional species, including dogs, horses, sea turtles, rabbits, koalas, guinea pigs, and members of the Dasypodidae family, such as the nine-banded armadillo [12,21,27,28,29,30,31,32,33,34], a detailed anatomic description of those images of the normal six-banded armadillo is lacking. Therefore, the present study was conducted to shed light on the anatomical details of the six-banded armadillo’s nasal cavity using anatomical cross-sectional images and their correlation with transverse CT images.

In this study, the combination of images acquired via different CT windows and planes and anatomical cross-sections were adequate for studying, in detail, the normal six-banded armadillos’ nasal cavities and paranasal sinuses without repositioning the animals. Hence, the anatomical sections provided precise anatomic characteristics of the rostral part of their head structures. Additionally, the CT images avoided the superimposition of bones, nasal turbinates, and teeth, facilitating a more general view of the entire nasal cavity. To the best of the authors’ knowledge, this is the first description of this region in the six-banded armadillo using anatomical cross-sections and conventional CT equipment. Similar studies accomplished on a variety of species, such as rhinoceros iguanas [12], horses [27,29], dogs [28,34], koalas [31], rabbits [32], and guinea pigs [33], have effectively demonstrated that the combination of advanced imaging techniques and anatomical cross-sections is adequate for studying the complex anatomy of this region. 

Considering the nine-banded armadillo, different structures were identified in the transverse plane, which is the plane most frequently used in veterinary medicine [34]. Although the transverse CT images provided remarkable anatomical information, the combination of the transverse, sagittal, and dorsal planes made the identification of some structures more obvious. Thus, the lung and bone tissue CT windows helped distinguish the main formations of the nasal cavity, such as the dorsal, ventral, middle, and common nasal meatuses. In addition, the bone CT window showed excellent details of the dorsal conchal sinus, and the different parts of the nasal ventral conchae. As observed in the rabbit, the dorsal nasal concha narrowed rostrally, and widened gradually to form a large dorsal conchal sinus. Thus, our results were similar to those observed in this and other species [27,32,33,34]. In addition, the presence of air in other cavities was quite helpful in providing good contrast in the CT images, helping to identify the oral cavity, paranasal sinuses, and some of their specific communications due to the different attenuations among structures. Concerning the different paranasal sinuses identified, they were mostly represented by the maxillary sinuses. Similar findings have already been reported in other exotic species, such as the guinea pig [33]. The evaluation of the nasal and paranasal sinuses along with the cheek teeth is pivotal for the detection of a variety of processes [33] that are common in lagomorphs and rodents [35], such as the malocclusion of the dental structures, dental abnormalities, and dental abscesses. These diseases may lead to infections that extend to these sinuses and produce rhinosinusitis [36]. Therefore, a CT of the head for reasons other than studies of the nasal cavity is a helpful tool for use on pet rabbits and rodents to diagnose the presence and the extent of abscesses, as well as the interpretation of different pathological conditions of the rostral part of the armadillo’s head using CT scans. 

On the other hand, several studies performed on exotic species have been achieved with micro-CT equipment, as it allows for a better distinction of anatomical formations compared to conventional CT scans due to their intrinsic properties [8,21]. However, the use of this specific equipment is sparse in veterinary hospitals. Despite this limitation, the implementation of different CT windows and anatomical cross-sections used in this study provided notable knowledge for anatomic interpretation in teaching and clinical practice.

## 5. Conclusions

This research is the first description of a six-banded armadillo’s nasal cavity and paranasal sinuses using transverse, sagittal, and dorsal CT images in combination with anatomical cross-sections. The images obtained in this study were quite helpful in identifying the anatomical landmarks of this region. Moreover, these techniques could facilitate teaching applied anatomy to our students as these procedures contribute to the visualization of structures without overlapping, eliminating the difficulties in visualizing the organization of the rostral part of the six-banded armadillo’s head. 

## Figures and Tables

**Figure 1 animals-14-01135-f001:**
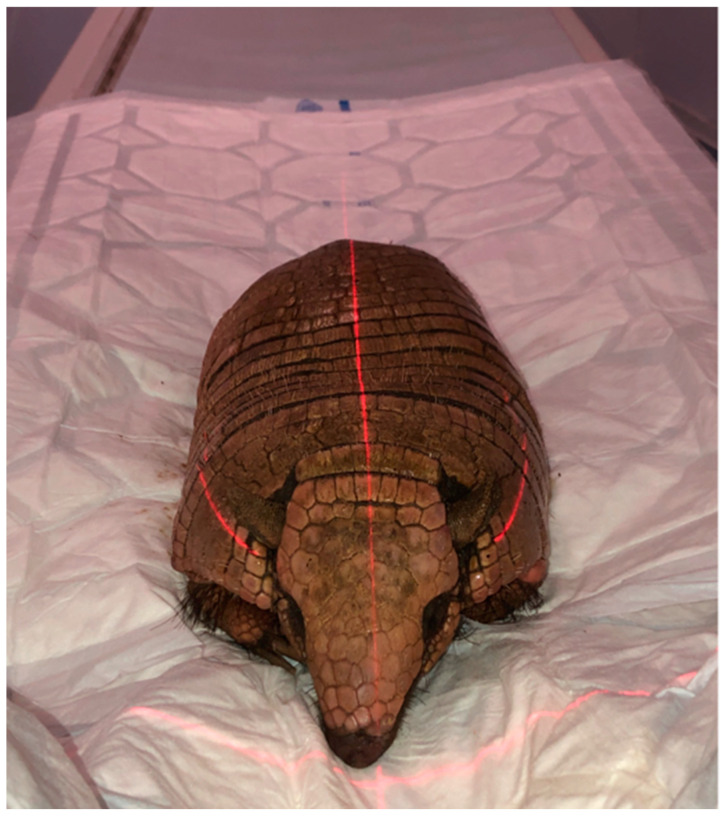
Image of a six-banded armadillo on the CT scan table.

**Figure 2 animals-14-01135-f002:**
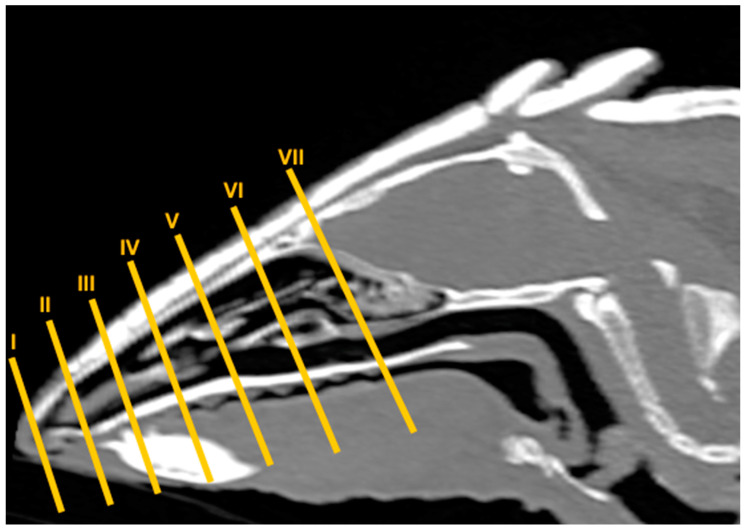
Parasagittal MPR CT image of the head of a six-banded armadillo representing the approximate level of the slices of this study. Segments I–VII correspond to Figure 2, Figure 3, Figure 4, Figure 5, Figure 6, Figure 7, Figure 8 and Figure 9.

## Data Availability

The information is available at “https://accedacris.ulpgc.es”.

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
