# Peer review of "Correlation between Cross-Sectional Anatomy and Computed Tomography of the Normal Six-Banded Armadillo (Euphractus sexcintus) Nasal Cavity and Paranasal Sinuses"

_animals, 2024, doi:10.3390/ani14071135_

Round 1
Reviewer 1 Report
Comments and Suggestions for Authors
Dear Editor,
The current study investigated the nasal cavity of six banned armadillo by cross CT and correlated to sectional anatomy.
The study is interesting and results are clear.
I have few points to report.
- Avoid using "we, our"
- replace madibular incisors by lower incisors.
- Fif 8. Mg is it mylohyoid ms or geniohyoid ms?
Authors did not provide adequate information about the paranasal sinuses and their divisions and communications both in results and in discussion. They should discuss similarities and differences with other species.
Authors should provide gross images for paranasal sinuses
Author Response
Dear Reviewer,
We really appreciate all your comments since they have been quite helpful in improving our manuscript.
Following your suggestions, we have done the changes as you recommend.
- Therefore, in the new version of the manuscript we have avoided the use of "we or our".
- In the same way, we have replaced "madibular incisors" by "lower incisors".
- Fig. 8. Mg is it mylohyoid ms or geniohyoid ms. It is a quite interesting question, since the mylohyid ms forms the floor of the oral cavity. Thus, we have included both in the figure as "Mg+Mh".
- "Authors did not provide adequate information about the paranasal sinuses and their divisions and communications both in results and in discussion. They should discuss similarities and differences with other species." As you suggest, we have included some divisions and communicatios of the paranasal sinuses that were clearly visible, such as those of the maxillary sinus. This information and similarities with related species is presented in the discussion section.
- Authors should provide gross images for paranasal sinuses. We agree with the reviewer, nonetheless, due to the low number of specimens used in this study, we focused on obtaining transverse section with the adequate quality since they allowed us to obtain more information of this region (see discussion section).
Reviewer 2 Report
Comments and Suggestions for Authors
L90-91 The objectives are different from the data presented in the conclusions. L97-98 Exclude. L107-108 This comment is part of the results. L153 Adjust according to the NAV. L178 Exclude.
Figures 1-10 insert a scale bar.
What is the main question addressed by the research? Propose a comparative assessment between anatomy and CT.
What specific gap in the field does the paper address?High-quality images ensuring the possibility to enhance the understanding of species specifications.
Are the references appropriate? YES
Please include any additional comments on the tables and figures and quality of the data. NO

Author Response
Dear Reviewer,
We have added your suggestions in order to improve the quality of our manuscript.
Thus, L90-91 The objectives are different from the data presented in the conclusions. Following your recommendation, these sentences have been deleted.
L97-98 Exclude. These sentences have been excluded as you suggested.
L 107-108, No anatomic variations were detected in the head of the six-banded armadillos used in the investigation. As you recommend, we have moved these sentences to results section.
L153 Adjust according to the NAV. We have revised it according to the Nomina Anatomica Veterinaria (see text).
L178 Exclude. As you recommend, we have deleted it.
Figures 1-10 insert a scale bar. In the revised version of the manuscript, we have inserted a scale bar.
In addition, we have deleted the sentences highlighted in the conclusion section.
Round 2
Reviewer 1 Report
Comments and Suggestions for Authors
I think the manuscript is suitable for publication